# The underappreciated role of agricultural soil nitrogen oxide emissions in ozone pollution regulation in North China

Xiao Lu[1,2], Xingpei Ye [1], Mi Zhou[1], Yuanhong Zhao[3], Hongjian Weng[1], Hao Kong[1], Ke Li [4], Meng Gao[5], Bo Zheng [6], Jintai Lin [1], Feng Zhou [7], Qiang Zhang [8], Dianming Wu [9], Lin Zhang [1✉] & Yuanhang Zhang[10✉]

Intensive agricultural activities in the North China Plain (NCP) lead to substantial emissions of nitrogen oxides ($NO_x$) from soil, while the role of this source on local severe ozone pollution is unknown. Here we use a mechanistic parameterization of soil $NO_x$ emissions combined with two atmospheric chemistry models to investigate the issue. We find that the presence of soil $NO_x$ emissions in the NCP significantly reduces the sensitivity of ozone to anthropogenic emissions. The maximum ozone air quality improvements in July 2017, as can be achieved by controlling all domestic anthropogenic emissions of air pollutants, decrease by 30% due to the presence of soil $NO_x$. This effect causes an emission control penalty such that large additional emission reductions are required to achieve ozone regulation targets. As $NO_x$ emissions from fuel combustion are being controlled, the soil emission penalty would become increasingly prominent and shall be considered in emission control strategies.

[1] Laboratory for Climate and Ocean-Atmosphere Studies, Department of Atmospheric and Oceanic Sciences, School of Physics, Peking University, Beijing, China. [2] School of Atmospheric Sciences, Sun Yat-sen University, Zhuhai, Guangdong, China. [3] College of Oceanic and Atmospheric Sciences, Ocean University of China, Qingdao, China. [4] John A. Paulson School of Engineering and Applied Sciences, Harvard University, Cambridge, MA, USA. [5] Department of Geography, Hong Kong Baptist University, Hong Kong, China. [6] Laboratoire des Sciences du Climat et de l'Environnement, CEA-CNRS-UVSQ, Gif-sur-Yvette, France. [7] Laboratory for Earth Surface Processes, College of Urban and Environmental Sciences, Peking University, Beijing, China. [8] Ministry of Education Key Laboratory for Earth System Modeling, Department of Earth System Science, Tsinghua University, Beijing, China. [9] Key Laboratory of Geographic Information Sciences, School of Geographic Sciences, East China Normal University, Shanghai, China. [10] State Key Joint Laboratory of Environmental Simulation and Pollution Control, College of Environmental Sciences and Engineering, Peking University, Beijing, China. ✉email: zhanglg@pku.edu.cn; yhzhang@pku.edu.cn

Surface ozone is a major air pollutant that is harmful to human health and vegetation[1–3]. Extensive surface ozone measurements from global monitoring networks have revealed that summertime ozone levels, and the associated health exposures over the North China Plain (NCP) are significantly higher than those over other northern mid-latitude regions[4,5]. Despite the fact that the Chinese Action Plan on Air Pollution Prevention and Control implemented in 2013 has significantly reduced the nationwide anthropogenic emissions of primary pollutants including particulate matter (PM) and nitrogen oxides ($NO_x = NO + NO_2$)[6,7], summertime ozone pollution, measured as daily 8 h average maximum (MDA8) has been increasing at over 3 ppbv year$^{-1}$ in the NCP over 2013–2019, among the fastest urban ozone trends in the recent decade reported in the Tropospheric Ozone Assessment Report (TOAR)[8–10]. Recent studies suggested that the ozone increases were likely driven by decreases of PM and anthropogenic $NO_x$, and changes in meteorological conditions[11–15]. The observed ozone increases during the coronavirus disease 2019 (COVID-19) lockdown in China also reflected the complexity of ozone mitigation[16–18]. Here we show that the substantial soil $NO_x$ emissions present an additional challenge for ozone pollution regulation in the NCP.

Surface ozone is primarily produced from the sunlight-driven oxidation of volatile organic compounds (VOCs) and carbon monoxide (CO) in the presence of $NO_x$. These precursors are emitted from both anthropogenic (fuel combustion from power plants, industry, transportation, and residential sources) and biogenic sources (e.g., $NO_x$ from soil). Being the most intensive anthropogenic emission regions in China[19], the NCP also contains 23% of Chinese cropland areas (agricultural areas of about 300,000 km$^2$) and uses 30% of the national fertilizer consumption[20]. The intensive nitrogen inputs to soil from fertilizer applications[21] and nitrogen deposition[22] lead to large soil $NO_x$ emissions via microbial processes reaching 20% of the anthropogenic $NO_x$ emissions in summer over the NCP[13,23,24]. The soil $NO_x$ emissions from both the natural nitrogen pool and fertilizer input are conventionally considered as biogenic sources, and are not considered in the current design of emission control strategies in China[7,25].

The contribution of soil $NO_x$ emissions to ozone formation in the NCP is complicated by the nonlinear ozone chemistry in the presence of high anthropogenic sources. The efficiency of ozone formation largely depends on the photochemical regime, i.e., whether it is sensitive to $NO_x$ ($NO_x$-limited regime) or VOCs ($NO_x$-saturated regime) or both (transitional regime). Observational and modelling studies have shown that ozone formation in the NCP is typically in transitional or $NO_x$-saturated regime in urban and suburban areas, and in $NO_x$-limited regime for rural areas[11,26–29]. Significant ozone enhancements from agricultural soil $NO_x$ emissions in $NO_x$-limited regions were suggested in some recent studies[30–33], yet no studies so far have examined how soil $NO_x$ emissions interact with anthropogenic sources in $NO_x$-rich regions such as the NCP. Soil $NO_x$ emissions are typically simplified or neglected in many air quality models applied for ozone source attributions and emission control strategy assessments in China[15,34,35], and the implication of this missing source is still unknown.

In this work, we address the issue by applying two atmospheric chemistry model simulations (GEOS-Chem and WRF-Chem) under different anthropogenic and soil emission scenarios. Soil $NO_x$ emissions are estimated by a mechanistic parameterization and can be supported by field measurements and satellite observations of tropospheric $NO_2$ columns. We demonstrate that the presence of soil $NO_x$ emissions in the NCP that largely driven by fertilizer application, significantly reduces the sensitivity of surface ozone to anthropogenic $NO_x$ emissions, degrades the effectiveness of anthropogenic emissions control measures on surface ozone regulation, and therefore serves as a penalty requiring extra anthropogenic emission reduction. This study highlights the previously underappreciated important role of soil $NO_x$ emissions on accurate attribution of anthropogenic ozone sources that is crucial for designing ozone pollution regulation strategies.

## Results and discussion

**Anthropogenic and soil $NO_x$ emissions in the NCP.** Figure 1 compares the anthropogenic and soil $NO_x$ emissions over China at 0.25° × 0.3125° resolution in July 2017. We choose July here as it is a typical boreal summer month with intensive soil emissions and severe ozone pollution in the NCP[13]. Anthropogenic $NO_x$ emissions from the Multi-resolution Emission Inventory for China (MEIC[6,19]; with latest available year 2017) include combustion sources, i.e., industry, transportation, power plant, and residential processes, while agricultural $NO_x$ emissions are not included. The total combustion induced anthropogenic $NO_x$ emissions over China in July 2017 are 0.53 Tg N, with 31% (0.16 Tg N) of them emitted in the NCP region. The MEIC inventory estimates that anthropogenic $NO_x$ emissions in the NCP for July peaked at 0.23 Tg N in 2011 and has decreased since then due to emission control measures[6], consistent with trends in satellite observed $NO_2$ tropospheric columns[36].

Soil $NO_x$ emissions are calculated using the Berkeley-Dalhousie Soil $NO_x$ Parameterization (BDSNP) as a function of available soil nitrogen content from fertilizer application and nitrogen deposition, and edaphic conditions such as soil moisture and temperature[37,38]. Its implementation in the GEOS-Chem model driven by assimilated meteorological fields allows the online calculation of hourly soil $NO_x$ emissions at each model grid (Methods; Supplementary information). The estimated annual total soil $NO_x$ emissions above canopy in 2008–2017 are 0.77 ± 0.04 Tg N per annum (Tg N a$^{-1}$) (mean ± standard deviation of annual totals) in China and 0.18 ± 0.01 Tg N a$^{-1}$ in the NCP with small meteorology-driven interannual variability. A distinct seasonal variation exists in the soil $NO_x$ emissions, with the highest emissions of 0.034 ± 0.003 Tg N month$^{-1}$ in May–July 2008–2017 and 0.03 Tg N in July 2017 in the NCP (Supplementary Fig. 1). Separating soil $NO_x$ from natural soil nitrogen content and fertilizer content in the BDSNP parameterization indicates that fertilizer-induced emissions (Methods) are the main component in eastern China, accounting for 58% of the July soil $NO_x$ emissions in the NCP (Fig. 1c). Compared with MEIC, the soil $NO_x$ emissions in the NCP are about 11–20% of the anthropogenic sources in July 2008–2017, and become higher in more recent years due to the decline of the latter. From a global perspective, the NCP stands out with both high anthropogenic and soil $NO_x$ emissions, in contrast to other surface ozone hot spots such as the US, Europe, Japan, and Korea recorded in TOAR[9,10] where the two sources are typically well separated spatially (Supplementary Fig. 2).

Our estimated soil $NO_x$ emissions above canopy of 0.77 ± 0.04 Tg N a$^{-1}$ in China are comparable with previous studies in the range of 0.4–1.3 Tg N a$^{-1}$, and consistent with independent field measurements across China (Fig. 1d, Supplementary Tables 1 and 2). The NCP is a region with intensive croplands (Supplementary Fig. 1), with high surface $NO_x$ and $N_2O$ concentrations being observed in the region after fertilizer applications[39–41]. The presence of soil $NO_x$ emissions in the NCP is further evident from satellite observations of tropospheric $NO_2$ column. We compare in Fig. 1e and Supplementary Fig. 3 the GEOS-Chem model simulated tropospheric $NO_2$ columns with or without soil $NO_x$ emissions to three OMI $NO_2$ satellite products (Methods). Simulated tropospheric $NO_2$ columns in the NCP with soil

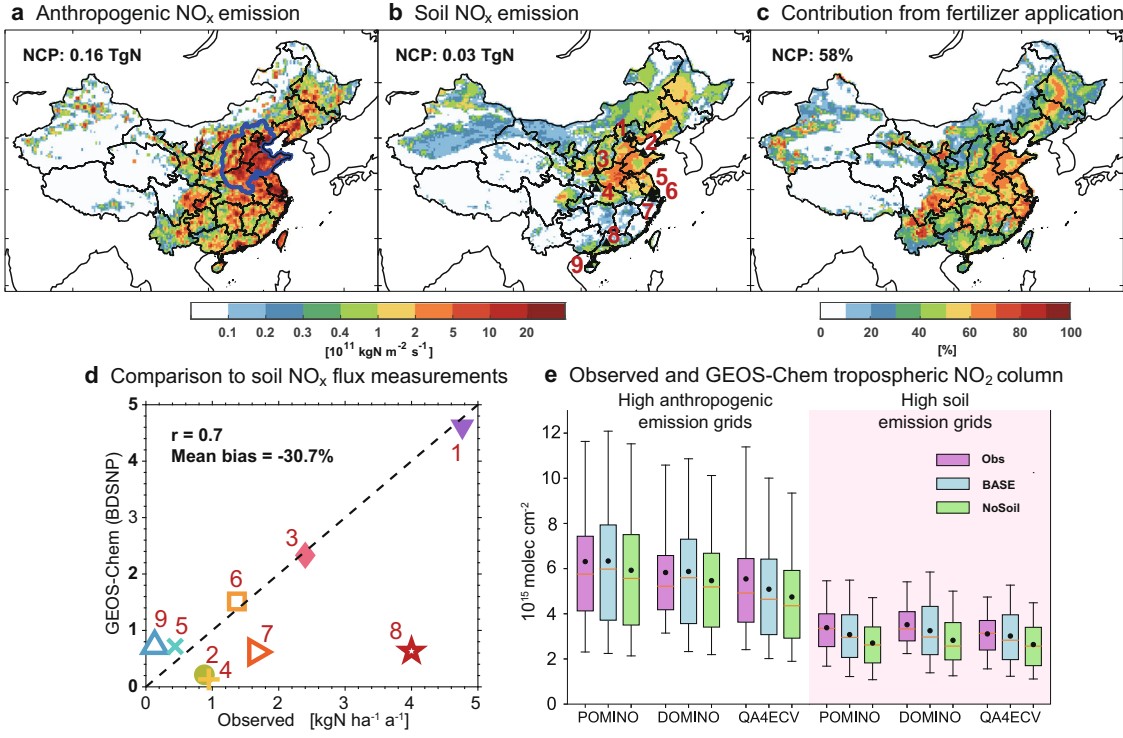

**Fig. 1 Substantial anthropogenic and soil NOₓ emissions lead to high NO₂ levels over the North China Plain (NCP).** Panels **a** and **b** show the anthropogenic NOₓ emissions in July 2017 from the Multi-resolution Emission Inventory for China (MEIC) and the soil NOₓ emissions calculated from the Berkeley-Dalhousie Soil NOₓ Parameterization (BDSNP) implemented GEOS-Chem, respectively. The thick blue lines outline the NCP region. The total emissions in the NCP are shown in the inset. Panel **c** shows the fraction of soil NOₓ emissions from fertilizer application to the total soil NOₓ emissions. Panel **d** compares the BDSNP soil NOₓ emissions to nine field measurements across China (locations given in the Panel **b** and Supplementary Table 2, with the correlation coefficient and mean bias shown in the inset. Panel **e** compares the GEOS-Chem simulated tropospheric $NO_2$ columns over the NCP with averaging kernel applied to the POMINO, DOMINO, and QA4ECV satellite products (Methods). The colored box-and-whisker plots (5th, 25th, 50th, 75th, and 95th percentiles, and mean values denoted as dots) represent $NO_2$ columns from the observation, GEOS-Chem BASE simulation, and a sensitivity model simulation with soil NOₓ emissions excluded (NoSoil). The comparisons are grouped for the high anthropogenic NOₓ emission model grids (defined as grids with the 20% anthropogenic/soil NOₓ emission ratio greater than 2, accounting for 20% of the NCP grids), and high soil NOₓ emission model grids (defined as grids with the 20% anthropogenic/soil NOₓ emission ratio smaller than 0.5, accounting for 30% of the NCP grids) (Supplementary Fig. 1c). We use the emission ratio of 20% as the criteria here as the July soil NOₓ emissions in the NCP are about 20% of the anthropogenic NOₓ emissions (**a, b**). Supplementary Figure 3 compares the spatial distributions.

emissions are consistent with the observations with mean differences less than 5%, but if soil NOₓ emissions are excluded model results would be biased low by 15–20% ($P < 0.01$) in areas with low anthropogenic/soil emission ratios (Supplementary Fig. 1c), and by 12–14% ($P < 0.01$) for all the NCP areas.

**Impact of soil NOₓ emissions on ozone formation in the NCP.** We analyze how soil NOₓ emissions affect ozone formation from anthropogenic sources in the NCP region. Ozone enhancements from a specific source can be determined in atmospheric chemistry models as the differences between the standard simulation with all emissions turned on and a sensitivity simulation with this source turned off or perturbed (Methods; Supplementary Table 3). Here we apply the GEOS-Chem chemical transport model with the MEIC anthropogenic emissions and BDSNP soil NOₓ emissions at 0.25° × 0.3125° resolution over China (Methods). Our previous work has evaluated the GEOS-Chem ozone simulation for March–October 2016–2017 with the same model configuration using measurements from the nationwide monitoring network of the Chinese Ministry of Ecology and Environment[13]. We show in Fig. 2 and Supplementary Fig. 4 that the model reproduces the spatial pattern of ozone distribution ($r = 0.72$), with a small positive mean bias of 2 ppbv for MDA8 ozone measured at the NCP cities in July 2017.

Substantial differences are found in anthropogenic ozone enhancements simulated by turning off domestic anthropogenic sources in the presence vs. absence of soil NOₓ emissions (Fig. 2c and d). Monthly mean anthropogenic ozone enhancements in July average 21.2 ppbv in the NCP when soil NOₓ emissions are considered (16.6–24.8 ppbv with a factor of 2 uncertainty in soil NOₓ emissions, i.e., by applying 200% or 50% of the BDSNP-estimated Chinese soil NOₓ emissions in the model as informed by Supplementary Table 1), which is 30% (19–46%) lower than the value of 30.7 ppbv if soil NOₓ emissions are removed in GEOS-Chem model simulations. These anthropogenic ozone values estimate the largest ozone reduction that can be achieved by controlling domestic anthropogenic emissions of air pollutants, and thus are crucial for assessing the effectiveness and potential of emission control measures. The large 9.5 ppbv mean differences reflect a strong interactional effect of domestic anthropogenic emissions with soil NOₓ emissions in the NCP that has not been studied before. Additional analyses on July 2016 and 2018 suggest that this effect is robust for other years with small interannual variabilities in the magnitude (Supplementary Fig. 5).

We find a similar strong effect of soil NOₓ emissions on anthropogenic ozone in the NCP using the WRF-Chem regional air quality model, suggesting the feature is robust among air

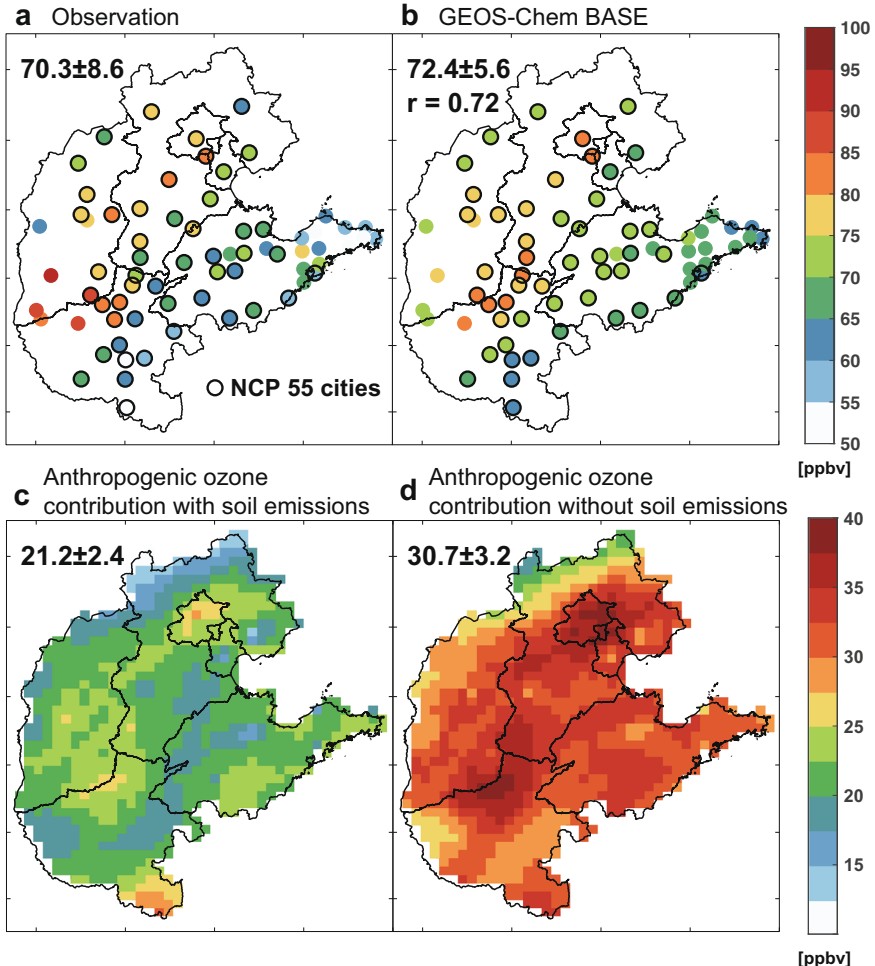

**Fig. 2 Soil NO$_x$ emission influences on surface ozone levels and estimated anthropogenic ozone contribution.** Panels **a** and **b** show the spatial distribution of **a** observed and **b** simulated mean MDA8 ozone at urban sites over the NCP in July 2017. Mean values ± standard deviation and their spatial correlation coefficients (*r*) in the 55 NCP cities categorized as key cities for air pollution monitoring (marked with black circles) are shown in the inset. Panels **c, d** show ozone contributions from domestic anthropogenic emissions, estimated as the simulated ozone difference with domestic anthropogenic sources turned on/off, in the presence/absence of soil NO$_x$ emissions (Methods). The mean values ± standard deviation over the NCP grids are shown in the inset.

quality models. We apply the same anthropogenic and soil NO$_x$ emissions in the WRF-Chem model (Methods) as those used in GEOS-Chem (Supplementary Fig. 6). The mean anthropogenic ozone enhancement in the NCP estimated by WRF-Chem is 40 (60) ppbv in the presence (absence) of soil NO$_x$ emissions (Supplementary Fig. 7). The WRF-Chem model estimates larger anthropogenic ozone enhancements than GEOS-Chem, likely caused by the different treatments of other background sources and chemical mechanisms. However, the two models agree on the 30% reduction of anthropogenic ozone estimates when soil NO$_x$ emissions are considered in the NCP.

The presence of soil NO$_x$ emissions appears to suppress the sensitivity of summertime ozone pollution to anthropogenic sources in the NCP. This effect is different from another natural NO$_x$ source from lightning that emits in the free troposphere and has a much smaller influence on anthropogenic ozone attribution at the surface (Supplementary Fig. 8). We further illustrate the effects by conducting a series of GEOS-Chem sensitivity simulations for July 2017 with Chinese domestic anthropogenic NO$_x$ emissions reduced by, respectively, 20%, 40%, 60%, 80%, and 100% with and without soil NO$_x$ sources (Methods; Supplementary Table 3). Figure 3 shows the evolution of the NCP July mean surface MDA8 ozone concentrations under the

different anthropogenic emission reduction scenarios. As we gradually reduce anthropogenic NO$_x$ emissions, ozone concentrations would decrease at accelerating rates, suggesting increasing efficiency of NO$_x$ control measures. The suppressed sensitivity of ozone to anthropogenic NO$_x$ imposed by soil NO$_x$ emissions, as indicated by the difference between the ozone decrease rates with same anthropogenic NO$_x$ reduction, in the presence vs. absence of soil emissions, also become greater under larger emission reduction conditions.

We use the ratio of surface H$_2$O$_2$ to HNO$_3$ concentrations (hereafter H$_2$O$_2$/HNO$_3$) as an indicator of the ozone formation regime[42,43]. Although the threshold of H$_2$O$_2$/HNO$_3$ for determining ozone formation regime varies regionally[42], a higher H$_2$O$_2$/HNO$_3$ value typically indicates ozone formation being more sensitive to NO$_x$ emissions. As seen in Fig. 3, the NCP mean H$_2$O$_2$/HNO$_3$ ratio is only 0.2 under the base condition, indicating a NO$_x$-saturated or transitional ozone formation regime, consistent with previous observation-based or model-inferred estimates[26,27]. H$_2$O$_2$/HNO$_3$ values increase with decreasing anthropogenic NO$_x$ emissions, and become greater than 1 for scenarios with over 80% NO$_x$ emission reductions. If soil NO$_x$ emissions were excluded, the ozone formation regime in the NCP would shift towards a more NO$_x$-sensitive condition at the same

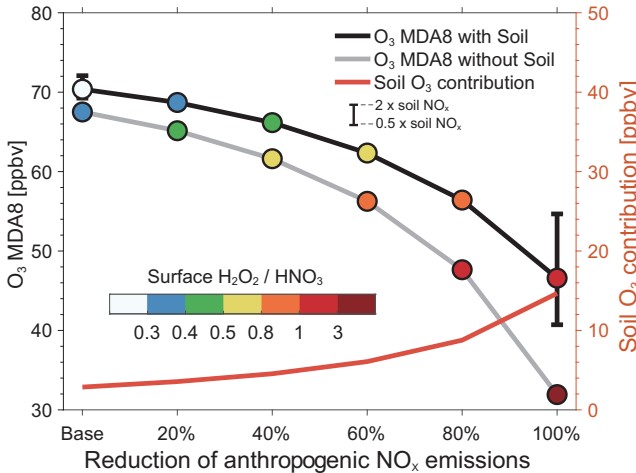

**Fig. 3 The presence of soil NOₓ emissions suppresses the sensitivity of ozone to anthropogenic NOₓ emissions.** The GEOS-Chem simulated responses of MDA8 ozone in the NCP to the decline of anthropogenic NOₓ emissions by 20, 40, 60, 80, and 100% relative to July 2017 levels (left y-axis), in the presence (black line) and absence (gray line) of soil NOₓ emissions. The black bars represent the range of estimates with a factor of 2 uncertainty in soil NOₓ emissions. The colored circles represent the mean surface H₂O₂/HNO₃ ratios for each emission scenario. The red line (right y-axis) shows the corresponding ozone contribution from soil NOₓ emissions as can also be estimated from the difference between the black and gray lines.

anthropogenic NOₓ levels, and the shift would become much more distinct when anthropogenic NOₓ emissions are largely controlled and the soil becomes an increasingly important source of NOₓ.

The interactional effect of soil and anthropogenic NOₓ emissions is also reflected by the suppression of ozone produced from soil NOₓ at high anthropogenic NOₓ levels. Ozone produced from soil emissions estimated from excluding soil NOₓ emissions in a sensitivity simulation (Methods) is only 2.9 ppbv (1.7–4.6 ppbv considering a factor of 2 uncertainty in soil NOₓ emissions) in the NCP relative to the base simulation, and would increase to 15 ppbv (9–23 ppbv) when all domestic anthropogenic NOₓ emissions were turned off (Fig. 3). The more NOₓ-sensitive chemical regime with reducing anthropogenic NOₓ emissions enhances the ozone production efficiency and thus leads to a greater ozone contribution from the soil. The enhanced soil ozone contribution offsets the expected ozone decrease driven by reduced anthropogenic NOₓ ozone contribution alone (gray line in Fig. 3), explaining the smaller total ozone reduction with vs. without soil NOₓ emissions.

We also find different influences of anthropogenic and soil NOₓ emissions on ozone formation in the NCP. Removing soil NOₓ emissions lower the July mean surface ozone levels by 2.9 ppbv, while reduction of a similar NOₓ amount from anthropogenic sources (i.e., 20% of anthropogenic sources as shown in Fig. 1) in the region would lead to 1.7 ppbv lower ozone with a different spatial pattern (Supplementary Fig. 9) that can be largely attributed to the different spatial distribution of emissions. Reducing domestic anthropogenic NOₓ emissions by 20% would lead to ozone increases in cities, such as Beijing, Tianjin, and Shijiazhuang (Supplementary Fig. 9) as ozone formation in these urban areas is NOₓ-saturated due to high anthropogenic NOₓ emissions. By contrast, removing soil NOₓ emissions would in general decrease ozone concentrations in the NCP, in particular over the high soil NOₓ emissions areas where ozone formation is more sensitive to NOₓ. Ozone in these areas tends to have a

longer lifetime than that in urban areas due to less titration (Supplementary Fig. 9), and can thus transport long distances and compensate ozone changes in the NOₓ-saturated areas. This can be further supported by estimates of ozone production efficiency (OPE), defined as the number of ozone molecules produced per molecule of NOₓ emitted[43]. The NCP mean OPE in July contributed by soil NOₓ emissions is 28% higher than that contributed by 20% anthropogenic NOₓ emissions in the base simulation (36.4% vs. 8.3% in Supplementary Fig. 9).

**The implication of soil NOₓ emissions on ozone mitigation strategies.** Our finding that soil NOₓ emissions strongly affect the sensitivity of ozone concentrations to anthropogenic sources in the NCP raises the need to assess its role in emission control strategies for improving ozone air quality. To address this issue, we apply GEOS-Chem model simulations to examine the responses of surface ozone in the NCP cities to reductions of anthropogenic NOₓ, VOCs, and CO emissions relative to the July 2017 level in the presence vs. absence of soil NOₓ emissions. We analyze three ozone exposure metrics relevant to air quality, human and vegetation health: MDA8, NDGT70, and AOT40[5,44] (Methods). Figure 4 shows the predicted percentage changes of these ozone metrics at the 55 NCP cities (Fig. 2a) under different emission reduction scenarios for July 2017. Compared to MDA8, the metrics of NDGT70 and AOT40 show greater sensitivity to declines in anthropogenic NOₓ emissions. It reflects high ozone concentrations as emphasized by the NDGT70 and AOT40 metrics are more sensitive to NOₓ emission reductions than ozone averages such as MDA8. Ozone formation in the NCP cities are typically NOₓ-saturated (Supplementary Fig. 10), so that joint reduction of anthropogenic NOₓ, VOCs, and CO emissions can be more effective for ozone air quality improvement than controlling NOₓ emissions alone for small reduction intensities, but further ozone improvement will be determined by NOₓ reduction as ozone formation shifts eventually to be NOₓ limited (Fig. 4).

Distinct differences can be seen for the predicted ozone changes with vs. without soil NOₓ emissions considered (Fig. 4, Supplementary Table 4). Predicted ozone decreases associated with emission reductions would be much faster for all three metrics if there were no-soil NOₓ emissions. For 20% anthropogenic NOₓ emission reduction, predicted July mean ozone decreases in the NCP are 1.4% for MDA8, 2.3% for AOT40, and 4.6% for NDGT70 with soil NOₓ emissions included in the simulations. However, these values are 2.6% for MDA8, 5.5% for AOT40, and 12% for NDGT70 without soil NOₓ emissions considered. In case of a greater emission reduction such as 60%, predicted ozone decreases are 8.8% for MDA8, 18% of AOT40, and 34% for NDGT70 in the presence of soil NOx emissions vs. 15% for MDA8, 33% of AOT40, and 58% for NDGT70 in the absence of soil NOₓ emissions.

The 2018–2020 Chinese Clean Air Action plan called for a 9% decrease for NOₓ emissions and 10% for VOC emissions relative to the 2017 level[25]. These reductions are very likely not strong enough to reduce ozone levels at the NCP cities as seen from our model projection and from observations[8,12], and more stringent emission control measures on NOₓ and VOCs are under design[45]. We show that large soil NOₓ emissions in the NCP present a previously overlooked challenge for future emission controls. To quantify it, we define the "soil NOₓ penalty" as the extra required anthropogenic emission reduction to achieve a target ozone level compared to the condition without soil NOₓ emissions. As shown in Fig. 4c, d, to achieve 5 ppbv reduction of MDA8 ozone on the basis of July 2017 conditions (~70 ppbv), 41% reduction of domestic anthropogenic NOₓ emissions as estimated from a

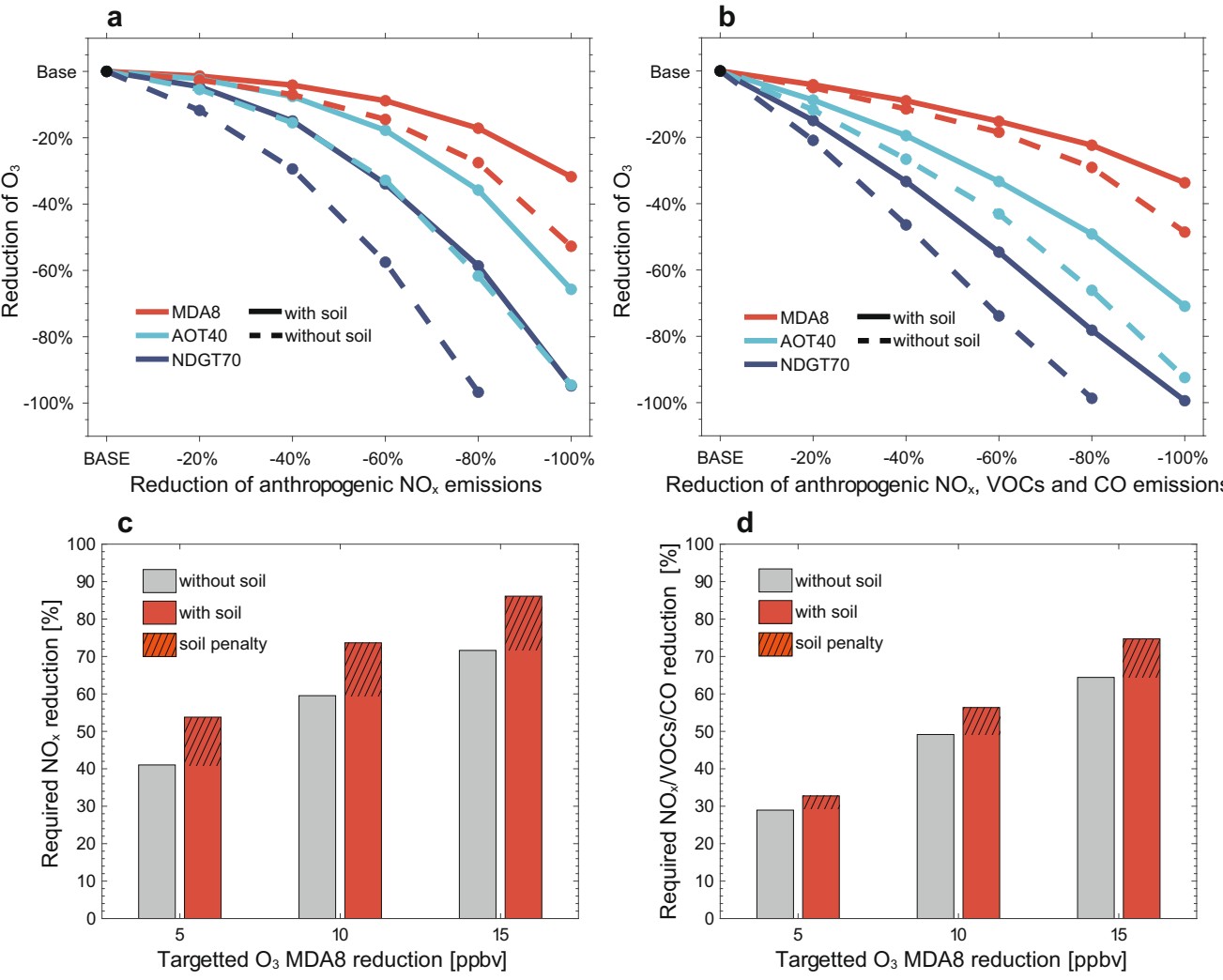

**Fig. 4 The "soil NO$_x$ emission penalty" on anthropogenic ozone pollution regulation averaged in the 55 NCP cities.** Panels **a** and **b** show the responses of ozone metrics (MDA8, AOT40, and NDGT70, Methods) to the reductions of anthropogenic NO$_x$ emissions, and the joint reductions of anthropogenic NO$_x$, VOCs, and CO emissions, respectively. Panels **c** and **d** show the required reduction of anthropogenic NO$_x$ emissions or joint reduction of anthropogenic NO$_x$, VOCs, and CO emissions for different ozone control targets in the NCP cities, estimated from panels **a** and **b** by a logarithmic fitting (Supplementary Table 5), both in the presence (gray) and absence (red) of soil NO$_x$ emissions. The difference between the red and gray bars thus illustrates the soil NO$_x$ emission penalty, i.e., extra anthropogenic emission reduction imposed by the presence of high soil NO$_x$ emissions.

logarithmic fitting function (Supplementary Table 5) would be required if there were no-soil NO$_x$ emissions (i.e., if models neglect soil NO$_x$ emissions), but additional 13% emission reduction is required if soil NO$_x$ emissions are accounted for. The soil NO$_x$ emission penalty increases to 15% for a more ambitious ozone reduction target of 15 ppbv. Jointly controlling anthropogenic NO$_x$, VOCs, and CO emissions by the same ratios can reduce the soil NO$_x$ penalty to 3.8 and 10% for 5 and 15 ppbv ozone reduction targets, respectively. It can be expected that ozone control strategy design based on predictions without considering soil NO$_x$ emissions would largely underestimate the emission control efforts required to achieve an ozone air quality target.

In summary, our analyses have revealed an underappreciated role of soil NO$_x$ emissions, largely caused by agricultural fertilizer applications, on ozone air quality in the NCP. Our model simulations indicate that although the presence of soil NO$_x$ emissions here may only enhance the mean ozone concentrations by 2.9 ppbv for July 2017, it significantly hampers surface ozone pollution regulation by suppressing the sensitivity of ozone to combustion induced anthropogenic NO$_x$ emissions. It leads to

additional 13–15% emission reductions (or 3.8–10% if also controlling VOCs and CO) required to achieve ozone pollution reductions of 5–15 ppbv in the NCP cities. As combustion induced anthropogenic NO$_x$ emissions are being gradually reduced, such soil NO$_x$ emissions penalties would become increasingly prominent, and thus shall be accounted for in emission control strategies. We call for more direct measurements of soil NO$_x$ to better constrain its emissions in this region. The soil is also an important source of nitrous acid (HONO)[46,47], another reactive nitrogen species contributing to ozone formation, and can have an even greater role than reported here in ozone air quality that needs to be assessed in future studies. The soil NO$_x$ effects on ozone air quality as revealed in this study can also be important in the Indo-Gangetic Plain, another region with high emissions of both anthropogenic and soil NO$_x$ (Supplementary Fig. 2). Management of the soil NO$_x$ emissions by improving the efficiency of nitrogen fertilizer application may have co-benefits on air quality, human health, food security, climate mitigation, and biodiversity conservation, helping solving the global nitrogen challenge[48].

## Methods

**Surface ozone observations over China.** Surface ozone observations over China in July 2017 were obtained from the China National Environmental Monitoring Center (CNEMC) network (http://106.37.208.233:20035/). The CNEMC network reports hourly surface ozone concentrations at over 1500 sites covering more than 450 cities. Ozone measurements were reported in units of μg m⁻³ at the standard atmospheric condition (273.15 K, 1 atm) and were converted to volume mixing ratios (ppb) in this study. We applied data quality control measures to remove unreliable data following our previous study[5].

**Ozone metrics relevant to air quality, human health, and vegetation exposure.** We analyzed three ozone metrics (MDA8, NDGT70, and AOT40) characterizing ozone pollution and its impacts on human health and vegetation, following the Tropospheric Ozone Assessment Report (TOAR)[44]. The daily 8 h average maximum ozone (MDA8) is the standard metric used for ozone air quality regulation in China, and is widely used in cohort studies examining the responses of human health to ozone exposure[2,49]. The number of days with MDA8 > 70 ppb (NDGT70) estimates the frequency of extreme ozone pollution episodes and acute health impact to ozone exposure[44]. The cumulative daytime hourly ozone concentrations of >40 ppb (AOT40) estimates ozone damages to vegetation.

**Satellite observations of tropospheric NO₂ column.** We used observations of tropospheric NO₂ column in July 2017 retrieved from the Ozone Monitoring Instrument (OMI). OMI is on board the NASA Earth Observing System (EOS) Aura satellite with an ascending equator crossing time at ~13:45 local time (LT). It measures backscattered solar radiation in the ultraviolet and visible wavelength range of 270–504 nm[50], and has a near-daily global coverage at a swath width of 2600 km and a pixel resolution of $13 \times 24$ km² at nadir view[45]. The OMI NO₂ observations have been extensively applied to monitor NO₂ air pollution and to interpret NOₓ emissions over China[24,26,36,51].

We obtained OMI tropospheric NO₂ columns from three retrievals: the Peking University POMINO level 2 product version 2[52,53] (https://www.amazon.com/clouddrive/share/4tTaCCGYblD17KpJjh4PNnsGOetqwFffyPEzQChoaKz), the Dutch OMI NO₂ level 2 product (DOMINO) version 2.0[54] (https://www.temis.nl/airpollution/no2.php), and the European Quality Assurance for Essential Climate Variables (QA4ECV) project's NO₂ ECV precursor level 1.1 product[55] (https://www.temis.nl/airpollution/no2.php). We excluded pixels with snow-covered surfaces, row anomaly, or cloud fractions higher than 30 %[36]. For comparison with the different observation products, GEOS-Chem simulated NO₂ mixing ratios at 13–14 LT were sampled along the satellite tracks and smoothed by the corresponding averaging kernels.

**Soil NOₓ emissions in China estimated by the Berkeley-Dalhousie Soil NOₓ Parameterization (BDSNP) and from the literature.** The soil NOₓ emissions were estimated using the BDSNP[37] implemented in GEOS-Chem. Meteorological variables used in the BDSNP scheme are obtained from the GEOS-FP assimilated meteorological data, available hourly at a horizontal resolution of 0.25° (latitude) × 0.3125° (longitude). The soil NOₓ emissions were calculated at each model grid and each hour. Here we briefly summarized the key features in the BDSNP parameterization, and more information could be found in the Supplement and from Hudman et al. (2012)[37].

The BDSNP parameterizes global soil NOₓ emissions ($Emis_{soil}$) as a function of available soil nitrogen content, climate, and edaphic conditions following:

$$Emis_{soil} = A'_{biome}(N_{avail}) \times f(T) \times g(\theta) \times P(l_{dry}) \quad (1)$$

where $N_{avail}$ represents available soil nitrogen mass, $A'_{biome}$ denotes the biome-dependent emission factors, $f(T)$ and $g(\theta)$ are the temperature and soil moisture dependences, and $P(l_{dry})$ describes the pulsed soil emissions from wetting of dry soils[37]. The soil temperature and moisture term $f(T) \times g(\theta)$ is given as:

$$f(T) \times g(\theta) = e^{0.103T} \times a\theta e^{-b\theta^2} \quad (2)$$

where $T$ ($0 \le T \le 30°$) is the soil temperature and $\theta$ ($0 \le \theta \le 1$) is the water-filled pore space. The Poisson function $g(\theta)$ describes the dependence on soil moisture. $\theta$ is defined as the ratio of the volumetric soil moisture content to the porosity. It is available hourly from the GEOS-FP meteorological fields for the top 2 cm of soil, where the majority of the soil NOₓ release. The values of $a$ and $b$ are chosen such that $g(\theta)$ maximizes when $\theta = 0.2$ for arid soils and $\theta = 0.3$ elsewhere according to laboratory and field measurements[37].

The pulsing term $P(l_{dry})$ describes the pulsing of soil NOₓ emissions from a reactivation of water-stressed bacteria when very dry soil is wetted due to irrigation and/or rainfalls. It follows Yan et al. (2005)[56,57] and is given as:

$$P(l_{dry}) = [13.01 \ln(l_{dry}) - 53.6] \times e^{-ct} \quad (3)$$

where $l_{dry}$ is the length of the antecedent dry period in hours, $c$ is a constant rate denoting the rise/fall time of the pulse, and $t$ is the model emission time step.

The BDSNP considers available soil nitrogen content ($N_{avail}$) from the natural pool, fertilizer application, and nitrogen deposition. Fertilizer applications are obtained from the global gridded chemical fertilizer and manure application

inventory at $0.5° \times 0.5°$[56,57], in which the chemical fertilizers were spatially disaggregated from the International Fertilizer Association (IFA) national totals for year 2000 conditions, and the manure fertilizer were obtained from the Food and Agriculture Organization of the United Nations (FAO) Gridded Livestock of the World (GLW) project. We find that the Chinese chemical fertilizer application (straight N application) from IFA used in this study gives 19.6 Tg N a⁻¹ for 2000, comparable to the estimate of 22.2 Tg N a⁻¹ for 2017 from the China Statistical Yearbook (http://www.stats.gov.cn/tjsj/ndsj/). The uncertainties in the fertilizer input can be considered in our sensitivity simulations with different soil NOₓ scenarios. The annual fertilizer applications are then distributed over the satellite-derived growing season at each grid. The $N_{avail}$ from dry and wet nitrogen deposition is available from GEOS-Chem for each time step and is thus coupled to the model chemistry and deposition of reactive nitrogen compounds.

BDSNP estimated the multi-year mean global soil NOₓ emissions above canopy of 8.8–9.5 Tg N a⁻¹ using the MERRA2 (0.5° × 0.625°, 1980–2017) or GEOS-FP (0.25° × 0.3125°, 2014–2017) assimilated meteorological fields, within the range reported in previous bottom-up (3.3–10 Tg N a⁻¹) and top-down (7.9–16.8 Tg N a⁻¹) estimates[58]. Here we summarized the estimated soil NOₓ emission over China at domestic or regional scales from previous studies in Supplementary Table 1, adapted from Huang et al. (2014)[59]. Three approaches were typically used, including (1) statistical or mechanistic models using meteorological parameters and edaphic conditions (e.g., soil temperature and moisture) to parameterize soil NOₓ emissions, based on field measured relationships between these variables and soil NOₓ emissions; (2) top-down estimates using satellite NO₂ observations with the a priori inventory to constrain soil NOₓ emissions; and (3) upscaling the measurements from field campaigns to develop soil NOₓ emission inventories at regional or national scales. We found that the estimated Chinese annual soil NOₓ emissions above canopy from these studies showed a wide range of 0.48–1.38 Tg N a⁻¹. A recent study combining modeling and measurements from Huang et al. (2014)[59] estimated the soil NOₓ emissions of 1.23 (95% Confidence Limit: 0.59–2.13) Tg N year⁻¹. Our estimates of 0.77 Tg N a⁻¹ using the BDSNP parameterization were in the middle of the range. BDSNP-estimated soil NOₓ fluxes were also comparable to field measurements across China (Fig. 1d, Supplementary Table 2). The wide range of soil NOₓ emission estimates reflected the differences in the methods and location/time focuses among these studies, and also the uncertainties in the BDSNP parameterization and/or in meteorological fields and fertilizer application input. We thus conducted sensitivity simulations by assuming a factor of 2 uncertainty (i.e., by applying 50% or 200% of the BDSNP estimates in the model as informed by Supplementary Table 1) in the BDSNP-estimated Chinese soil NOₓ emissions (Supplementary Table 3) and to quantify how the uncertainty in the soil NOₓ emission affects our analyses.

**GEOS-Chem model simulation.** We used the global chemical transport model GEOS-Chem v11-02rc (http://geos-chem.org) to interpret surface ozone pollution and its source attribution over China in July 2017. The model simulates a state-of-the-art tropospheric HOₓ-NOₓ-VOCs-ozone-halogen-aerosol chemistry[60,61], and is driven by the GEOS-FP assimilated meteorological data from the Goddard Earth Observing System (GEOS) of the NASA Global Modeling and Assimilation Office (GMAO). The temporal resolution is 1 h for surface meteorological variables (including variables used in the BDSNP scheme) and boundary layer height and 3 h for others. We applied a nested version of the model over East Asia (70°–140°E, 15°–55°N) at a horizontal resolution of 0.25° (latitude) × 0.3125° (longitude). Boundary conditions were archived from the global simulation at 2° × 2.5° horizontal resolution. The model simulation covered July 2017 with a 6-month spin-up run for initialization.

Our GEOS-Chem simulations applied the latest Chinese anthropogenic emission inventory for 2017 from the Multi-resolution Emission Inventory for China (MEIC; http://www.meicmodel.org)[6]. The model also implemented a number of natural/biogenic emissions. Soil NOₓ emissions were calculated using the Berkeley-Dalhousie Soil NOₓ Parameterization (BDSNP) as introduced above. Lightning NOₓ emissions were parameterized as a function of cloud-top height and spatially constrained by satellite observations of lightning flash rates[62]. Biogenic VOC emissions were estimated by the Model of Emissions of Gases and Aerosols from Nature (MEGAN version v2.1) algorithm[63]. Biomass burning emissions were from the Global Fire Emissions Database version 4 (GFED4)[64]. A more detailed model description and configuration can be found in Lu et al. (2019)[13].

We designed a BASE simulation and a total of 33 sensitivity simulations to examine the role of anthropogenic and natural/biogenic (including soil and lightning) NOₓ emissions in ozone source attribution. The standard simulation (BASE) applied the anthropogenic and natural/biogenic emissions as described above. We then assessed the ozone source attribution using the standard "brute-forced" zero-out approach. Sensitivity simulations were conducted by excluding anthropogenic emissions of all primary emitted species over China (NoAnthro), excluding soil NOₓ emissions (NoSoil), excluding lighting NOₓ emission (NoLight), and excluding the combinations of these emissions (NoSoilLight, NoAnthroSoil, NoAnthroLight, and NoAnthroSoilLight) (Supplementary Table 3). Ozone differences between these simulations were used to estimate the ozone contributions from anthropogenic and individual natural/biogenic NOₓ sources, and the interactional (nonlinear) effects between the sources. For instance, ozone differences between the BASE and NoAnthro (BASE−NoAnthro), and between the NoSoil and NoAnthroSoil (NoSoil−NoAnthroSoil), estimated the domestic

anthropogenic ozone enhancements in the presence/absence of soil $NO_x$ emissions, respectively. The comparison of BASE–NoAnthro and NoSoil–NoAnthroSoil illustrated how soil $NO_x$ emissions influenced anthropogenic ozone enhancements through the interactional effect with anthropogenic sources.

Six sensitivity simulations were conducted to examine the influences of uncertainties in the soil $NO_x$ emission on ozone formation and source attribution. This is done by applying 200% or 50% of the BDSNP-estimated Chinese soil $NO_x$ emissions in the model (i.e., a factor of 2 uncertainty to capture the range of soil $NO_x$ emission estimates from Supplementary Table 1) with three different anthropogenic emission scenarios (Supplementary Table 3).

We further examined the impacts of soil $NO_x$ emissions on the effectiveness of emission controls for ozone pollution mitigation. This was conducted by reducing the anthropogenic $NO_x$ emissions over China by 20, 40, 60, 80, and 100% relative to 2017 levels in BASE, both in the presence and absence of soil $NO_x$ emissions. We repeated these sets of simulations but with joint reductions of anthropogenic $NO_x$, NMVOCs, and CO emissions over China (Supplementary Table 3).

**WRF-Chem model simulation**. We applied the WRF-Chem model version 3.6.1 with online-coupled meteorology and chemistry[65]. The modeling domain of WRF-Chem covered eastern China with a 27 km horizontal resolution and 37 vertical layers. The initial and lateral boundary conditions of meteorology were provided by National Centers for Environmental Prediction (NCEP) FNL (Final) Operational Global Analysis data at 1° × 1° horizontal resolution. The chemical conditions, physical and chemical parameterization schemes are summarized in Supplementary Table 6.

Our standard WRF-Chem simulation for July 2017 (WRF-Chem BASE) applied the same MEIC inventory for anthropogenic emissions over China as used in GEOS-Chem. It also implemented the MEGAN version 2.1 for biogenic VOCs emissions. MEGAN version 2.1 estimated soil $NO_x$ emissions as a function of temperature but the emissions were significantly underestimated as found in a number of previous studies[32,33,63]. Our WRF-Chem BASE simulation thus excluded soil $NO_x$ emissions calculated from MEGAN version 2.1. We then applied the monthly mean soil $NO_x$ emissions for July 2017 over China archived from the GEOS-Chem simulation as offline soil $NO_x$ emissions in the improved WRF-Chem simulation (WRF-Chem BASE + Soil). We further conducted WRF-Chem simulations with Chinese anthropogenic emissions excluded from the BASE and BASE + Soil scenarios, in order to evaluate the role of soil $NO_x$ emissions in ozone source attribution over China from the WRF-Chem model.

## Data availability

Surface ozone measurements in China are available at http://106.37.208.233:20035. Satellite observations of tropospheric $NO_2$ column are available at https://www.amazon.com/clouddrive/share/4tTaCCGYblD17KpJjh4PNnsGOetqwFffyPEzQChoaKz (the POMINO product), https://www.temis.nl/airpollution/no2.php (the DOMINO version 2.0 product), and https://www.temis.nl/airpollution/no2.php (the QA4ECV product). Observations of soil $NO_x$ emissions are available from the references as listed in Supplementary Table 1. Modeling outputs and data generated in this study have been deposited in https://doi.org/10.5281/zenodo.4740433[66] and is publically available.

## Code availability

GEOS-Chem model codes, including the Berkeley-Dalhousie Soil $NO_x$ Parameterization (BDSNP), are available at https://github.com/geoschem/geos-chem/releases/tag/v11-02-rc. WRF-Chem model codes are available at https://github.com/wrf-model/WRF/releases/tag/V3.6.1.

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

## Acknowledgements

The work was supported by the National Key Research and Development Program of China (2017YFC0210102) and the National Natural Science Foundation of China (41922037 and 71961137011).

## Author contributions

L.Z., Y.H. Zhang, and X.L. designed the study. X.L., X.P.Y. and M.Z. performed model simulations and conducted data analysis with the assistance from Y.H. Zhao, H.J.W., H.K., K.L. and M.G., B.Z. and Q.Z. provided the Chinese anthropogenic emissions inventories. J.T.L. provided the POMINO $NO_2$ products. F.Z. provided the agricultural $N_2O$ emissions. D.W. provided the soil $NO_x$ flux measurements (Beijing, Shandong, and Hubei). X.L. and L.Z. wrote the paper with valuable inputs from all authors.

## Competing interests

The authors declare no competing interests.
