## [Peer Review File · Nature Communications]

REVIEWER COMMENTS

Reviewer #1 (Remarks to the Author):

This is an important issue. NO emissions from agricultural fields play an important part in ozone formation over North China Plain. In these intensively used agricultural areas with multiple harvests per year and excessive N use, and with soil and climatic conditions that may favour high NO emissions, large pulses of NO may lead to ozone formation. The authors have set up an interesting study, and I was especially interested in the biogenic soil NO emissions. The authors use the BDSNP model as described very briefly in lines 336-348. Since there was not enough information to understand the model concepts and the units and temporal resolution, I read the original paper by Hudman et al. (2012). There I found what actually the temperature and soil moisture dependencies are and the way the model simulates the short-term pulses. However, it is not clear how the authors calculate for example water-filled pore space and at what temporal resolution. These parameters are essential, and it is completely unclear how this was implemented in the model. In no part of the paper it is explained what the temporal resolution is, although the emissions are expressed in terms of $\text{kn/m}^2/\text{s}$ in Figure 1. I tried to find out, and the only reference to temporal resolution is in line 436 (monthly mean emission). So I am not sure I understand how a model that can describe short-term emission pulses, is used here. Also, I wonder how fertilizer inputs are simulated in the case of multiple cropping. Are these pulses not essential for determining the resulting tropospheric chemistry, if emissions may occur after one rainfall event, or irrigation? Irrigation is a very important practice in this region. NO emissions are particularly high after wetting of the soil, but if soils are continuously wet, this effect may disappear. I therefore wonder if the authors have considered this effect wetting-drying after irrigation/rainfall events correctly. This is a serious concern, and the authors do not clarify how they simulated this effect. In addition, I miss a validation of the BDSNP model results. If soil NO flux measurements are available, this can be done easily and will also give some more confidence about the implementation of the NO pulsing in the model. In summary, in the model description for soil NO emissions there are many unclarities regarding the temporal resolution, the pulsing, irrigation and double cropping.

Regarding the data used, I wonder about the data for fertilizer use. Both FAO and IFA statistics indicate that fertilizer use in China around 2010-2013 was much higher than what is in the text based on Chinese statistics. Even if FAO and IFA are wrong, this should be mentioned because it may change the picture a lot.

In the text the authors refer to NO emissions from fertilized fields as being natural. I think this should be biogenic, because there is not much natural about tilled heavily fertilized irrigated crop fields.
Lex Bouwman

Reviewer #2 (Remarks to the Author):

Report on the manuscript titled "The underappreciated role of agricultural soil nitrogen oxides emissions in ozone pollution regulation in North China"

Major claims of the paper:

In order to mitigate ozone in North China Plain, emissions of NO_x, one of the main precursors of ozone, from fuel combustion have been reduced following the 2018–2020 Chinese Clean Air Action plan. However, emissions of NO_x from soil, due to intensive agricultural activities with use of fertilizers, have not been taken into consideration in the Chinese Clean Air Action plan. The current manuscript shows that the role of emissions of NO_x from soil is not negligible. The authors talk about a soil emission penalty and show that it will become increasingly prominent along with the control and decrease of NO_x emissions from fuel combustion.

The major claims of the paper are novel and they will be of interest to others in the community and the wider field:

The mitigation of ozone is incredibly complex due to its non-linear chemistry. Recent studies on the impact of the lockdown in China and the rest of the world faced due to COVID-19, show that ozone rose even if emissions of ozone precursors from traffic and manufacturing sectors decreased (Le et al., 2020; Shi et al., 2021). Even though it is now understood that all sectors should be considered when controlling emissions of ozone precursors in order to improve air quality, NO_x emissions from soil are rarely mentioned. The environmental impact of NO_x emissions from soil is an issue at global scale (Houlton et al., 2019) but is not often discussed with a link to ozone distribution and trends. This study will be of particular interest for the second Tropospheric Ozone Assessment Report that will be focusing on the role of ozone precursors on surface ozone and its impact on health and vegetation (<https://igacproject.org/activities/TOAR/TOAR-II>).

The work is convincing:

The authors used in situ observations of ozone, tropospheric column of NO₂ from satellite, two atmospheric chemistry models. They detailed very well the emissions inventories and the characteristics of the models such as the mechanistic parametrization used to estimate the soil NO_x emissions.

Using a nested version over East Asia of a global model as well as a regional model to test the robustness of the results is particularly appreciated. The authors thoroughly made sure the inputs were consistent between the two models so the output are comparable.

The authors gave detailed information on their method so a researcher can reproduce the work.

References:

T.Le, Y.Wang, L.Liu, J.Yang, Y.L.Yung, G.Li, J.H.Seinfeld, Unexpected air pollution with marked emission reductions during the COVID-19 outbreak in China. *Science* 369, 702–706 (2020)

Shi Z, Song C, Liu B, Lu G, Xu J, Van Vu T, Elliott RJR, Li W, Bloss WJ, Harrison RM. Abrupt but smaller than expected changes in surface air quality attributable to COVID-19 lockdowns. *Sci Adv.* 2021 Jan 13;7(3):eabd6696. doi: 10.1126/sciadv.abd6696. PMID: 33523881; PMCID: PMC7806219.

Houlton BZ, Almaraz M, Aneja V, Austin AT, Bai E, Cassman KG, Compton JE, Davidson EA, Erisman JW, Galloway JN, Gu B, Yao G, Martinelli LA, Scow K, Schlesinger WH, Tomich TP, Wang C, Zhang X. A world of co-benefits: Solving the global nitrogen challenge. *Earths Future.* 2019;7:1-8. doi: 10.1029/2019EF001222. PMID: 31501769; PMCID: PMC6733275.

Reviewer #3 (Remarks to the Author):

This paper looks at the effect of NO_x emissions on ozone pollution in the North China Plane, focussing on July 2017 using a variety of emission inventories, soil emission models and chemistry modelling. The presence of soil NO_x emissions shifts ozone formation towards more NO_x-saturated conditions, and significantly reduces the sensitivity of ozone to anthropogenic emissions. The paper very much focusses on the North China Plane and includes very little comparison between other parts of the world.

The claims in the paper are novel, they will help in policy decisions, realizing the missing link in NO_x emissions and ozone control. They will be of interest to all air quality experts in China. The work is convincing but as a non-expert on the division of sources of NO_x, feel that they didn't justify enough why they chose the with or without 20 % anthropogenic NO_x emissions in many of the graphs. Just some references to whether these figures or ideas are used elsewhere in the world or if this is something characteristic for China.

The work could be reproduced using the same models and choosing another year (e.g. July 2018?). Focusing all on one month is justified but there could have been a brief comparison of how July could compare to January for example in terms of NO_x emissions and ozone.

The styles was clear and well justified and the figures were useful and clear.

The order of the paper was somewhat surprising, with the methods after the results, which made it end without any conclusions and since the abstract is not very detailed, a reader quickly reading the paper would find it hard to pull out the important conclusions. I suggest focusing attention on the conclusions.

Response to the reviewers' comments

Nature Communications manuscript NCOMMS-20-44819

Reviewer #1 Dr. Lex Bouwman (Remarks to the Author):

Comment [1-1]: This is an important issue. NO emissions from agricultural fields play an important part in ozone formation over North China Plain. In these intensively used agricultural areas with multiple harvests per year and excessive N use, and with soil and climatic conditions that may favour high NO emissions, large pulses of NO may lead to ozone formation. The authors have set up an interesting study, and I was especially interested in the biogenic soil NO emissions.

Response [1-1]: Thank you for the positive comments. Please find below our itemized responses. In particular, we have substantially improved the description of biogenic soil NO emissions in the model and added the comparison of modelled soil NO_x fluxes with available field measurements across China.

Comment [1-2]: The authors use the BDSNP model as described very briefly in lines 336-348. Since there was not enough information to understand the model concepts and the units and temporal resolution, I read the original paper by Hudman et al. (2012). There I found what actually the temperature and soil moisture dependencies are and the way the model simulates the short-term pulses. However, it is not clear how the authors calculate for example water-filled pore space and at what temporal resolution. These parameters are essential, and it is completely unclear how this was implemented in the model. In no part of the paper it is explained what the temporal resolution is, although the emissions are expressed in terms of kn/m²/s in Figure 1. I tried to find out, and the only reference to temporal resolution is in line 436 (monthly mean emission). So I am not sure I understand how a model that can describe short-term emission pulses, is used here.

Response [1-2]: Thank you for pointing it out. In the revised manuscript, we now write a much more detailed description of the BDSNP model parameterization in *Methods* and in *Supplement information*.

1) Regarding time resolution and short-term pulsing emissions, the time resolution of BDSNP depends on the driving GEOS-FP meteorological fields, which is hourly in this study. At each time step, the resulting soil NO_x flux will be fed into the model to proceed other processes (e.g. chemistry, deposition, and transport). As such, short-term pulsing emissions (calculated based on the soil moisture) and their impacts on air quality can be simulated in GEOS-Chem. We have added the following text to describe the BDSNP time resolution in the Section "Anthropogenic and soil NO_x emissions in the NCP":

"Soil NO_x emissions are calculated using the Berkeley-Dalhousie Soil NO_x Parameterization (BDSNP) as a function of available soil nitrogen content from fertilizer application and nitrogen deposition, and edaphic conditions such as soil moisture and temperature. Its implementation in the GEOS-Chem model driven by assimilated meteorological fields allows the on-line calculation of hourly soil NO_x emissions at each model grid (Methods; Supplementary information)."

We have also added the text below in *Supplement Information* on the pulsing emission (a brief version is also added in *Methods*):

“The pulsing term $P(l_{dry})$ describes the pulsing of soil NO_x emissions from a reactivation of water-stressed bacteria when very dry soil is wetted due to irrigation and/or rainfalls. It follows Yan et al. (2005) and is given as:

$$P(l_{dry}) = [13.01 \ln(l_{dry}) - 53.6] \times e^{-ct}, \quad (3)$$

where l_{dry} is the length of the antecedent dry period in hours which is updated in the model based on soil moisture from the meteorological fields, $c = 0.068 \text{ hour}^{-1}$ is a constant rate denoting the rise/fall time of the pulse, and t is the model emission time step.”

2) The water fill pore space θ in the model, defined as the ratio of the volumetric soil moisture content to the porosity, is indicated by soil moisture at the top 2 cm of soil where the majority of the soil NO_x release. Its hourly-average values are available from the GEOS-FP assimilated meteorological fields.

We now state in the *Methods*: *“The Poisson function $g(\theta)$ describes the dependence on soil moisture. θ is defined as the ratio of the volumetric soil moisture content to the porosity. It is available hourly from the GEOS-FP meteorological fields for the top 2 cm of soil, where the majority of the soil NO_x release. The values of a and b are chosen such that $g(\theta)$ maximizes when $\theta = 0.2$ for arid soils and $\theta = 0.3$ elsewhere according to laboratory and field measurements.”*

And also state in the *Supplement Information*:

“The term $g(\theta)$ describes the Poisson function scaling for soil moisture. θ is defined as the ratio of the volumetric soil moisture content to the porosity, and is normalized by dividing by the porosity so that $0 \leq \theta \leq 1$. It is available hourly from the GEOS-FP meteorological fields for the top 2 cm of soil, where the majority of the soil NO_x release. The values of a and b are chosen such that $g(\theta)$ maximizes when $\theta = 0.2$ for arid soils and $\theta = 0.3$ elsewhere according to laboratory and field measurements. As point out by Hudman et al. (2012), there is uncertainty on how well θ can reflect the real-world water-filled pore space, but the use of this parameter represents a mechanistic approach for soil NO_x emission estimates in the atmospheric chemical model that can take advantage of available assimilated meteorological fields.”

Comment [1-3]: Also, I wonder how fertilizer inputs are simulated in the case of multiple cropping. Are these pulses not essential for determining the resulting tropospheric chemistry, if emissions may occur after one rainfall event, or irrigation? Irrigation is a very important practice in this region. NO emissions are particularly high after wetting of the soil, but if soils are continuously wet, this effect may disappear. I therefore wonder if the authors have considered this effect wetting-drying after irrigation/rainfall events correctly. This is a serious concern, and the authors do not clarify how they simulated this effect.

Response [1-3]: The effect of soil wetting by rain or irrigation on soil NO_x emissions has been considered in the model as we described above. BDSNP uses hourly soil moisture from assimilated meteorological fields to update the length of antecedent dry period $P(l_{dry})$ in Equation (3) above, so that the high NO emissions after sudden wetting of soil and the decay of emissions due to continuous wetting can be simulated. Rainfalls or irrigation are reflected as changes of soil moisture in the meteorological field, and thus their effects on pulse soil NO_x are accounted for in our simulations.

Fertilizer nitrogen inputs are considered in the model (please also see Response [1-6] below), while multiple cropping or crop rotation is not simulated in the BDSNP. We have added the following text in *Supplement information* to describe the fertilizer inputs and applications: *“The annual fertilizer applications are then distributed over the satellite-derived growing season at each grid, with 75% of which are distributed over the first month as a Gaussian distribution around the green-up day, and the rest 25% are distributed evenly over the remaining growing season. Multiple cropping or crop rotations are not considered here.”*

Comment [1-4]: In addition, I miss a validation of the BDSNP model results. If soil NO flux measurements are available, this can be done easily and will also give some more confidence about the implementation of the NO pulsing in the model.

Response [1-4]: Thanks for pointing it out. We have gone through the literature and found nine field measurements of soil NO flux measurements in China. We now add Figure 1d and Supplementary Table 2 to show the comparison of BDSNP modelled soil NO_x fluxes to these field measurements. The results show that BDSNP generally captures the spatial pattern and magnitude of the measured soil NO_x fluxes across China, although with a mean negative bias, and support our application of BDSNP in the atmospheric chemical model to investigate the impacts of soil NO_x emissions on air quality.

We now state in the text *“Our estimated soil NO_x emissions above canopy of $0.77 \pm 0.04 \text{ Tg N a}^{-1}$ in China are comparable with previous studies in the range of $0.4\text{-}1.3 \text{ Tg N a}^{-1}$, and consistent with independent field measurements across China (Figure 1d, Supplementary Tables 1 and 2)”*.

Comment [1-5]: In summary, in the model description for soil NO emissions there are many unclarities regarding the temporal resolution, the pulsing, irrigation and double cropping.

Response [1-5]: Thank you for your comment. As described above, we have added the section “Supplementary information on the Berkeley-Dalhousie Soil NO_x Parameterization (BDSNP)” in *Supplement Information*, and also expanded the section in *Methods*. The new text now summarizes the features of BDSNP from Hudman et al. (2012), including its temporal resolution, the fertilizer application with cropping activities, and the dependences of pulsing emissions on rainfalls and irrigation. We think the section now gives a much clearer description on the BDSNP scheme. The newly added comparison of measured vs. BDSNP soil NO_x

fluxes further support the application of BDSNP over China.

Comment [1-6]: Regarding the data used, I wonder about the data for fertilizer use. Both FAO and IFA statistics indicate that fertilizer use in China around 2010-2013 was much higher than what is in the text based on Chinese statistics. Even if FAO and IFA are wrong, this should be mentioned because it may change the picture a lot.

Response [1-6]: Thank you for pointing it out. We indeed find that chemical fertilizer application (straight N application) in China shows a larger variation from the International Fertilizer Association (IFA, <https://www.ifastat.org/databases/plant-nutrition>) for 2000-2017, ranging from 29.5 Tg N a⁻¹ for 2009 to 10 Tg N a⁻¹ for 2017, compared to the 21-24 Tg N a⁻¹ for 2000-2017 estimated from the China Statistical Yearbook (CSY, <http://www.stats.gov.cn/tjsj/ndsj/>). However, the IFA estimation for 2000 (19.6 Tg N a⁻¹), which is used in this study, is comparable to the CSY estimation for 2017 (22.2 Tg N a⁻¹). We do not compare the grand total N because this information is not available from the CSY. We acknowledge that there might be uncertainties in the fertilizer input estimates, and our sensitivity simulations (with soil NO_x emissions increased by 100% or decreased by 50%) also reflect these uncertainties.

We have added the following text in *Supplement Information* (a brief version is presented in *Methods*): *“The BDSNP considers available soil nitrogen content from the natural pool, fertilizer application, and nitrogen deposition. Fertilizer applications are obtained from the global gridded chemical fertilizer and manure application inventory at 0.5° × 0.5° (Potter et al, 2010), in which the chemical fertilizers were spatially disaggregated from the International Fertilizer Association (IFA) national totals for year 2000 conditions, and the manure fertilizer were obtained from the Food and Agriculture Organization of the United Nations (FAO) Gridded Livestock of the World (GLW) project. We find that the Chinese chemical fertilizer application (straight N application) from IFA as used in this study gives 19.6 Tg N a⁻¹ for 2000, which is close to the estimate of 22.2 Tg N a⁻¹ for 2017 from the China Statistical Yearbook (<http://www.stats.gov.cn/tjsj/ndsj/>). The China Statistical Yearbook estimates relatively stable Chinese chemical fertilizer application of about 21-24 Tg N a⁻¹ in 2000-2017. The uncertainties in the fertilizer input can be considered in our sensitivity simulations with different soil NO_x scenarios.”*

Comment [1-7]: In the text the authors refer to NO emissions from fertilized fields as being natural. I think this should be biogenic, because there is not much natural about tilled heavily fertilized irrigated crop fields.

Response [1-7]: Thank you for the suggestion. We have changed the word “natural” to “biogenic” in the text.

References: Yan, X., Ohara, T. & Akimoto, H. Statistical modeling of global soil NO_x emissions. *Global Biogeochem. Cycles* 19, GB3019 (2005).

Hudman, R. C. *et al.* Steps towards a mechanistic model of global soil nitric oxide emissions: implementation and space based-constraints. *Atmos. Chem. Phys.* 12, 7779-7795 (2012).

Reviewer #2

Report on the manuscript titled “The underappreciated role of agricultural soil nitrogen oxides emissions in ozone pollution regulation in North China”

Comment [2-1]: Major claims of the paper: In order to mitigate ozone in North China Plain, emissions of NO_x, one of the main precursors of ozone, from fuel combustion have been reduced following the 2018–2020 Chinese Clean Air Action plan. However, emissions of NO_x from soil, due to intensive agricultural activities with use of fertilizers, have not been taken into consideration in the Chinese Clean Air Action plan. The current manuscript shows that the role of emissions of NO_x from soil is not negligible. The authors talk about a soil emission penalty and show that it will become increasingly prominent along with the control and decrease of NO_x emissions from fuel combustion.

The major claims of the paper are novel and they will be of interest to others in the community and the wider field:

The mitigation of ozone is incredibly complex due to its non-linear chemistry. Recent studies on the impact of the lockdown in China and the rest of the world faced due to COVID-19, show that ozone rose even if emissions of ozone precursors from traffic and manufacturing sectors decreased (Le et al., 2020; Shi et al., 2021). Even though it is now understood that all sectors should be considered when controlling emissions of ozone precursors in order to improve air quality, NO_x emissions from soil are rarely mentioned. The environmental impact of NO_x emissions from soil is an issue at global scale (Houlton et al., 2019) but is not often discussed with a link to ozone distribution and trends.

This study will be of particular interest for the second Tropospheric Ozone Assessment Report that will be focusing on the role of ozone precursors on surface ozone and its impact on health and vegetation (<https://igacproject.org/activities/TOAR/TOAR-II>).

Response [2-1]: Thank you for the positive comments and valuable references. We agree that ozone increases during the COVID-19 lockdown in China further illustrate the complexity for ozone mitigation. We now state in the text “*The observed ozone increases during the coronavirus disease 2019 (COVID-19) lockdown in China also reflected the complexity of ozone mitigation*”.

We also added the following text at the end of conclusion: “*Management of the soil NO_x emissions by improving the efficiency of nitrogen fertilizer application may have co-benefits on air quality, human health, food security, climate mitigation, and biodiversity conservation, helping solving the global nitrogen challenge (Houlton et al. 2019).*”

References:

- Houlton, B. Z., Almaraz, M., Aneja, V., Austin, A. T., Bai, E., Cassman, K. G., Compton, J. E., Davidson, E. A., Erisman, J. W., Galloway, J. N., Gu, B., Yao, G., Martinelli, L. A., Scow, K., Schlesinger, W. H., Tomich, T. P., Wang, C., and Zhang, X.: A world of co-benefits: Solving the global nitrogen challenge, *Earths Future*, 7, 1-8, <http://doi.org/10.1029/2019EF001222>, 2019.
- Le, T., Wang, Y., Liu, L., Yang, J., Yung, Y. L., Li, G., and Seinfeld, J. H.: Unexpected

air pollution with marked emission reductions during the COVID-19 outbreak in China, *Science*, 369, 702-706, <http://doi.org/10.1126/science.abb7431>, 2020.

Shi, Z., Song, C., Liu, B., Lu, G., Xu, J., Van Vu, T., Elliott, R. J. R., Li, W., Bloss, W. J., and Harrison, R. M.: Abrupt but smaller than expected changes in surface air quality attributable to COVID-19 lockdowns, *Sci Adv*, 7, <http://doi.org/10.1126/sciadv.abd6696>, 2021.

Comment [2-2]: The work is convincing:

The authors used in situ observations of ozone, tropospheric column of NO₂ from satellite, two atmospheric chemistry models. They detailed very well the emissions inventories and the characteristics of the models such as the mechanistic parametrization used to estimate the soil NO_x emissions.

Using a nested version over East Asia of a global model as well as a regional model to test the robustness of the results is particularly appreciated. The authors thoroughly made sure the inputs were consistent between the two models so the output are comparable.

The authors gave detailed information on their method so a researcher can reproduce the work.

Response [2-2]: Thank you for the positive comments. We have further improved our manuscript based on all the suggestions.

Reviewer #3 (Remarks to the Author):

Comment [3-1]: This paper looks at the effect of NO_x emissions on ozone pollution in the North China Plane, focussing on July 2017 using a variety of emission inventories, soil emission models and chemistry modelling. The presence of soil NO_x emissions shifts ozone formation towards more NO_x-saturated conditions, and significantly reduces the sensitivity of ozone to anthropogenic emissions. The paper very much focusses on the North China Plane and includes very little comparison between other parts of the world.

Response [3-1]: Thank you for the positive comments and careful edits. Please find below our itemized responses. We think the implication of this study is not only limited to the North China Plain (NCP), but also for other parts of the world with high emissions of both anthropogenic and soil NO_x, such as the Indo-Gangetic Plain. This soil NO_x penalty effect has not been reported in previous studies. We have now added the following text in the last paragraph: *“The soil NO_x effects on ozone air quality as revealed in this study can also be important in the Indo-Gangetic Plain, another region with high emissions of both anthropogenic and soil NO_x (Supplementary Figure 2).”*

Comment [3-2]: The claims in the paper are novel, they will help in policy decisions, realizing the missing link in NO_x emissions and ozone control. They will be of interest to all air quality experts in China. The work is convincing but as a non-expert on the division of sources of NO_x, feel that they didn't justify enough why they chose the with or without 20 % anthropogenic NO_x emissions in many of the graphs. Just some references to whether these figures or ideas are used elsewhere in the world or if this is something characteristic for China.

Response [3-2]: Thank you for pointing it out. Our choice of “20%” is from Figure 1, which shows that the total amount of soil NO_x emissions (0.03 Tg N) is about 20% of the anthropogenic NO_x emissions (0.16 Tg N) in the North China Plain in July, but with different spatial distributions. We thus use the 20% anthropogenic/soil NO_x emission ratio to group model grids (Fig.1 and Supplementary Figure 1c), and to compare the effects of reducing soil NO_x emissions vs. reducing 20% anthropogenic NO_x emissions on the NCP ozone air quality (Supplementary Fig. 9). The ratio of soil vs. anthropogenic NO_x emissions can be different for regions outside the NCP (as indicated by Supplementary Figure 2) but those regions are not the focus of this study.

We now state in the Section “Impact of soil NO_x emissions on ozone formation in the NCP”: *“Removing soil NO_x emissions lower the July mean surface ozone levels by 2.9 ppbv, while reduction of a similar NO_x amount from anthropogenic sources (i.e., 20% of anthropogenic sources as shown in Figure 1) in the region would lead to 1.7 ppbv lower ozone with a different spatial pattern (Supplementary Fig. 9) that can be largely attributed to the different spatial distribution of emissions.”*

We have also added the following text in the caption of Figure 1 and Supplementary Figures 1c and 9: *“We use the emission ratio of 20% as the criteria here as the July soil NO_x emissions in the NCP are about 20% of the anthropogenic*

NO_x emissions (Figs 1a and 1b).”

Comment [3-3]: The work could be reproduced using the same models and choosing another year (e.g. July 2018?). Focusing all on one month is justified but there could have been a brief comparison of how July could compare to January for example in terms of NO_x emissions and ozone.

Response [3-3]: Yes, we have conducted additional simulations for July 2016 and July 2018, and found that the soil NO_x effects on the model estimated anthropogenic ozone contribution are robust for different years. We do not conduct simulations for January because both ozone levels and soil NO_x emissions are low in the NCP or other regions of China during wintertime.

We have added the new Supplementary Figure 5, and now state in the text “*Additional analyses on July 2016 and 2018 suggest that this effect is robust for other years with small interannual variabilities in the magnitude (Supplementary Fig.5)*”.

Comment [3-4]: The styles was clear and well justified and the figures were useful and clear.

Response [3-4]: Thank you for the positive comments.

Comment [3-5]: The order of the paper was somewhat surprising, with the methods after the results, which made it end without any conclusions and since the abstract is not very detailed, a reader quickly reading the paper would find it hard to pull out the important conclusions. I suggest focusing attention on the conclusions.

Response [3-5]: We apologize for the confusion. However, the order/structure of this article follows the standard manuscript guideline of *Nature Communications* (<https://www.nature.com/documents/ncomms-submission-guide.pdf>). The guideline suggests the abstract to be a general introduction to the topic and a brief nontechnical summary of main results and implication with a word number limit of 150, as such the abstract may not include detailed descriptions. The guideline also requires the *Methods* Section to be placed after the *Results* Section, and does not suggest a separated *Conclusion* Section.

Comment [3-6]: Title: nitrogen oxide emissions on ozone pollution

Response [3-6]: We have changed the title to “*The underappreciated role of agricultural soil nitrogen oxide emissions in ozone pollution regulation in North China*”.

Comment [3-7]: GENERAL comment 1: You need to put the NCP throughout ALL the paper!! The North China Plane

Response [3-7]: Thanks for pointing it out. We have revised them as suggested.

Comment [3-8]: GENERAL comment 2: You need to change these in the figures and supplementary: shown in **the** inset

Response [3-8]: We have revised them as suggested.

Comment [3-9]: GENERAL comment 3: Your 20 % contribution of NO_x from soil feature throughout and you use this figure to include with or without but outside of China, I don't think the reader has enough scientific justification for these figures. Can you find some more international literature that justify this 20% or compare values in different parts of the world, so we can see how NCP compares to other regions around the world? Or is all derived or calculated from the numbers in the MEIC?

Response [3-9]: The 20% is derived from the comparison of MEIC anthropogenic emissions vs. BDSNP soil NO_x emissions over the NCP. Please see Response [3-2].

Comment [3-10]: Line 29 in **the** North China Plain (NCP) **has led to**

Response [3-10]: Revised as suggested.

Comment [3-11]: Line 30 the role of this source on **local** severe ozone pollution is unknown

Response [3-11]: Revised as suggested.

Comment [3-12]: Line 35: Not well written: Ozone air quality improvements were achieved/attained in July 2017 by controlling all domestic anthropogenic emissions and decrease by 30% due to soil NO_x emissions. (the 30% decrease is of the soil NO_x emissions or the domestic anthropogenic emissions or decrease of ozone? And when you talk about domestic anthropogenic emissions, maybe you should state what?)

Response [3-12]: We apologize for the confusion. We have now rephrased: *“The maximum ozone air quality improvements in July 2017, as can be achieved by controlling all domestic anthropogenic emissions of air pollutants, decrease by 30% due to the presence of soil NO_x.”*

Comment [3-13]: Line 38 isn't clear: If NO_x emissions from fuel combustion are controlled, the soil emission penalty would become increasingly prominent and shall be considered in emission control strategies.

Response [3-13]: We have rephrased the sentence to *“As NO_x emissions from fuel combustion are being controlled, the soil emission penalty would become increasingly prominent and shall be considered in emission control strategies.”*

Comment [3-14]: Line 42- Should this part be introduction?

Response [3-14]: Yes, we now add the heading of **“Introduction”** here.

Comment [3-15]: Line 43: Surface ozone

Response [3-15]: Revised as suggested.

Comment [3-16]: Line 49: particulate **matter** (PM)

Response [3-16]: Revised as suggested.

Comment [3-17]: Line 52: Reference 8 doesn't seem to be the TOAR assessment report, it may mention it but you should mention at least Fleming et al 2018 which talks about surface ozone: <https://online.ucpress.edu/elementa/article/doi/10.1525/elementa.273/112792/Tropospheric-Ozone-Assessment-Report-Present-day> You mention 34 Gaudel later, maybe you should mention that here too?

Response [3-17]: Thanks for pointing it out. We have added Fleming et al. (2018) and Gaudel et al. (2018) as references.

Comment [3-18]: Line 53: I think you should always say the NCP: 3 ppbv year⁻¹ in the NCP between 2013-2019

Response [3-18]: Revised as suggested.

Comment [3-19]: Line 61: The NCP also **contains** 23% of Chinese cropland areas (agricultural areas of about 300,000 km²) and **uses** 30% of the national fertilizer consumption

Response [3-19]: Revised as suggested.

Comment [3-20]: Line 65: from both **the** natural nitrogen pool and fertilizer input

Response [3-20]: Revised as suggested.

Comment [3-21]: Line 72: or both

Response [3-21]: Revised as suggested.

Comment [3-22]: Line 73: in **the** NCP is typically in the transitional or NO_x-saturated

Response [3-22]: Revised as suggested.

Comment [3-23]: Line 86 in **the** NCP that are largely driven by fertilizer

Response [3-23]: Revised as suggested.

Comment [3-24]: Line 97: Anthropogenic NO_x emissions from the Multi-resolution Emission Inventory for China (MEIC¹⁴; with latest available year 2017) include combustion sources, i.e., industry, transportation, power plant, and residential processes, while agricultural NO_x emissions are not included^{6,14}. I think you should reference 14

the first time you mention the model MEIC!

Response [3-24]: Thanks for pointing it out. We have revised it as suggested.

Comment [3-25]: Line 98: July peaked at 0.23 Tg N in 2011 and **has decreased** since then

Response [3-25]: Revised as suggested.

Comment [3-26]: Line 115: component in eastern China

Response [3-26]: Revised as suggested.

Comment [3-27]: Line 139: all **emissions** turned on

Response [3-27]: Revised as suggested.

Comment [3-28]: Line 152: 16.6-24.8 ppbv with a factor of 2 uncertainty in... (what is a factor of 2 uncertainty? A coverage factor, making it 95% significance?)

Response [3-28]: We have now rephrased the sentence: “... (16.6-24.8 ppbv with a factor of 2 uncertainty in soil NO_x emissions, i.e., by applying 200% or 50% of the BDSNP-estimated Chinese soil NO_x emissions in the model as informed by Supplementary Table 1) ...”. Please also see Response [3-33].

Comment [3-29]: Line 174: at the surface?

Response [3-29]: Revised as suggested.

Comment [3-30]:181-184: The sentence isn't clear: The suppressed sensitivity of ozone to anthropogenic NO_x imposed by soil NO_x emissions, as indicated by the difference between the ozone decrease rates with same anthropogenic NO_x reduction, in the presence vs. absence of soil emissions, also become greater under larger emission reduction conditions.

Response [3-30]: Thank you for the revision. We have revised it as suggested.

Comment [3-31]: Line 187: of **the** ozone formation..... for determining the ozone

Response [3-31]: Revised as suggested.

Comment [3-32]: Line 192: increase **with** decreasing anthropogenic

Response [3-32]: Revised as suggested.

Comment [3-33]: Line 202: factor of 2 uncertainty in soil NO_x emissions (does this mean it could be twice as high or half as high? I see in line 411 that you indeed look at

variations of 50 % or 200%. This is huge should you explain why the uncertainty is so high and what studies have been done to find it?

Response [3-33]: We have reviewed available studies on Chinese soil NO_x estimates as listed in Supplementary Table 1, and found that estimated Chinese annual soil NO_x emissions above canopy from different studies showed a range of 0.48-1.38 Tg N a⁻¹, compared to 0.77 Tg N a⁻¹ estimated from BDSNP used in this study. We therefore apply 50% or 200% of the BDSNP-estimated Chinese soil NO_x emissions in the model to capture the range of these studies and to test how the uncertainties may influence our result.

We have rephrased the text when we first mentioned the factor of 2 uncertainty (Response [3-28]), and also stated in *Methods* “*The wide range of soil NO_x emission estimates reflected the differences in the methods and location/time focuses among these studies, and also the uncertainties in the BDSNP parameterization and/or in meteorological fields and fertilizer application input. We thus conducted sensitivity simulations by assuming a factor of 2 uncertainty (i.e., by applying 50% or 200% of the BDSNP estimates in the model as informed by Supplementary Table 1) in the BDSNP estimated Chinese soil NO_x emissions (Supplementary Table 3) and to quantify how the uncertainty in the soil NO_x emission affects our analyses.*”

Comment [3-34]: Line 216: Reducing domestic anthropogenic NO_x emissions by **20%**

Response [3-34]: Revised as suggested.

Comment [3-35]: Line 222: long **distances** and ozone **molecules**

Response [3-35]: Revised as suggested.

Comment [3-36]: Line 226: This 20 % anthropogenic emissions you talk about in Fig 8- you have a case where it is eliminated, but it also seems to be added in the other cases and the NO_x emissions/soil NO_x emissions > 2 in Figure 8?! I cant directly see the 28 % in Fig 8 either.

Response [3-36]: The sensitivity simulations by reducing 20% anthropogenic emissions or reducing soil NO_x emissions apply to all model grids. The 28% is estimated as the difference between OPE enhancements, by reducing soil NO_x emissions (36.4% in Supp. Fig. 9g) vs. reducing 20% anthropogenic emissions (8.3% in Supp. Fig.9k). We have now rephrased the sentence: “*The NCP mean OPE in July contributed by soil NO_x emissions is 28% higher than that contributed by 20% anthropogenic NO_x emissions in the base simulation (36.4% vs. 8.3% in Supplementary Fig. 9).*”.

Comment [3-37]: Line 258: for a 9% decrease in NO_x emissions

Response [3-37]: Revised as suggested.

Comment [3-38]: Line 263: future emission **controls**

Response [3-38]: Revised as suggested.

Comment [3-39]: Line 266: basis of July 2017 conditions

Response [3-39]: Revised as suggested.

Comment [3-40]: Line 269-276: 13% emission reduction (32% extra effort) is required if soil NO_x emissions are accounted for. The soil NO_x emission penalty increases to 15% (20% extra effort) and later on. Could you explain the extra effort and how that differs from the No_x emission penalty?

Response [3-40]: We have deleted the “extra efforts” to avoid confusion.

Comment [3-41]: Line 285: NO_x emissions are gradually

Response [3-41]: We have rephrased the sentence: “As combustion induced anthropogenic NO_x emissions are being gradually reduced, ...”

Comment [3-42]: Line 299: in units of μg m⁻³ at

Response [3-42]: Revised as suggested.

Comment [3-43]: Line 368: affects our analyses

Response [3-43]: Revised as suggested.

Comment [3-44]: Line 373: state-of-the-art

Response [3-44]: Revised as suggested.

Comment [3-45]: Line 376: at a horizontal resolution of 0.25°

Response [3-45]: Revised as suggested.

Comment [3-46]: Line 387: Biogenic VOC emissions

Response [3-46]: Revised as suggested.

Comment [3-47]: Line 424: covered eastern China

Response [3-47]: Revised as suggested.

Comment [3-48]: You are lacking a conclusion. I would expect methods to come before results and “The implication of soil NO_x emissions on ozone mitigation strategies” to be a summary of the conclusions!!! But you really should have a conclusions section! So change the order of methods and results!

The titles are:

Results and Discussion

Anthropogenic and soil NO_x emissions in NCP.

Impact of soil NO_x emissions on ozone formation in NCP

The implication of soil NO_x emissions on ozone mitigation strategies

Methods

Surface ozone observations over China

Ozone metrics relevant to air quality, human health, and vegetation exposure

Satellite observations of tropospheric NO₂ column

Soil NO_x emissions in China estimated by the Berkeley-Dalhousie Soil NO_x Parameterization (BDSNP) and from the literature

GEOS-Chem model simulation

WRF-Chem model simulation

Response [3-48]: Thank you for the comment. Please see Response [3-5]. Following the journal manuscript guideline, we place the *Methods* section after *Results*. We also do not include a conclusions section, but conclude our findings and implications in the last paragraph of the manuscript.

Comment [3-49]: FIGURES

Line 627: are shown in the inset.

Some of the figures have the legends blurred!

Response [3-49]: We have revised accordingly and improved the resolution of the Figures. Some degradation of the resolution occurred when inserting them to the PDF document.

Comment [3-50]: Supplementary info

Line 45 black solid line

Line 63 are shown in **the** inset (In UK English anyway we say in the inset- have a look online and Line 68 a) and b) show

Line 69 are shown in the inset

Line 100 are shown in the inset

Line 102 panels (a)-(d), but show

Response [3-50]: The above comments have been revised as suggested.

REVIEWERS' COMMENTS

Reviewer #1 (Remarks to the Author):

Thank you for addressing my comments.
Lex Bouwman

Response to the reviewers' comments

Nature Communications manuscript NCOMMS-20-44819A

We thank Dr. Bouwman and the other two reviewers for the positive and valuable comments. The manuscript has been greatly improved since its initial submission. We believe this manuscript now provides convincing results revealing the important role of soil emissions of nitrogen oxides.

REVIEWERS' COMMENTS

Reviewer #1 (Remarks to the Author):

Thank you for addressing my comments.

Lex Bouwman